

# Research insights into the chemokine-like factor (CKLF)-like MARVEL transmembrane domain-containing family (CMTM): their roles in various tumors

Sai-Li Duan[1,2], Yingke Jiang[1], Guo-Qing Li[2], Weijie Fu[2], Zewen Song[3], Li-Nan Li[4] and Jia Li[4]

[1] Department of General Surgery, Xiangya Hospital Central South University, Changsha Province, Hunan, China
[2] Xiangya School of Medicine, Central South University, Changsha Province, Hunan, China
[3] Department of Oncology, The Third Xiangya Hospital of Central South University, Changsha Province, Hunan, China
[4] Department of Oncology, The 1st Affiliated Hospital of Dalian Medical University, Dalian, Liaoning, China

## ABSTRACT

The chemokine-like factor (CKLF)-like MARVEL transmembrane domain-containing (CMTM) family includes CMTM1–8 and CKLF, and they play key roles in the hematopoietic, immune, cardiovascular, and male reproductive systems, participating in the physiological functions, cancer, and other diseases associated with these systems. CMTM family members activate and chemoattract immune cells to affect the proliferation and invasion of tumor cells through a similar mechanism, the structural characteristics typical of chemokines and transmembrane 4 superfamily (TM4SF). In this review, we discuss each CMTM family member's chromosomal location, involved signaling pathways, expression patterns, and potential roles, and mechanisms of action in pancreatic, breast, gastric and liver cancers. Furthermore, we discuss several clinically applied tumor therapies targeted at the CMTM family, indicating that CMTM family members could be novel immune checkpoints and potential targets effective in tumor treatment.

## INTRODUCTION

The newly discovered CKLF-like MARVEL transmembrane domain-containing (CMTM) family comprises CMTM1–8 and CKLF, which are related to the chemokine family and the transmembrane 4 superfamily (TM4SF). Of these, the comparison between CKLF1, CMTM1 and CMTM2 and chemokine reveals a higher similarity, while CMTM8 exhibits a higher level of sequence homology with TM4SF11, reaching up to 39% at the overall amino acid level. The characteristics of the remaining members of the CMTM family exhibit an intermediate nature between CMTM1 and CMTM8 (*Duan et al., 2020*). The proteins encoded by the CMTM1–8 genes possess structural characteristics that are commonly

Corresponding authors
Li-Nan Li, lilinan200106@sina.com
Jia Li, lijia_211@hotmail.com

found in typical chemokines and TM4SF. CMTM family members have important roles in not only tumor pathogenesis but also reproductive and immune systems (*Wu et al., 2019*). They can activate and chemotactically attract immune cells, which impact tumor cell proliferation and invasion. This suggests that CMTM family members play a significant role in tumorigenesis and may be promising therapeutic targets (*Wu et al., 2020*). Studies have shown that CMTM expression is associated with pancreatic, breast, gastric, and liver cancer (*Li et al., 2014a*, *2020b*), and its mechanism of action is also related to gene methylation. Methylation of promoter CpG leads to silencing of the CMTM3 gene, which inhibits its function (*Wang et al., 2009*). Additionally, CMTM overexpression can activate the apoptosis pathway, which leads to induce tumor cell death in renal cancer carcinoma (*Wu et al., 2020*, *2022*). This review discusses the roles of CMTM family members in various cancers and the underlying mechanisms of action. Finally, we discuss the potential of the CMTM family members as therapeutic targets for tumors to improve patient outcomes.

## SURVEY METHODOLOGY

The existing literature was extensively investigated using Google Scholar and PubMed to analyze the role of CMTM families in different cancers, focusing on the mechanisms of action of different CMTM family members and several targeted therapies in clinical application in recent years. The scientific literature reviewed was not refined by journal type, author or date of publication. We also considered the previous published literature on CMTM cross-references to identify other appropriate relevant resources.

## CMTM FAMILY

The CMTM family consists of nine members, namely, CKLF and CMTM1-CMTM8, which are located on various chromosomes (Table 1). CMTM1–4 and CKLF are clustered on chromosome 16, while CMTM6–8 form gene clusters on chromosome 3. Additionally, CMTM5 is located on chromosome 14. Most of CMTM family members exhibit alternative RNA splicing events, resulting in the production of at least one splicing isoform containing MARVEL transmembrane domain protein (*Li et al., 2020b*). The functional attributes of CMTM gene products lie in an intermediate position between classical chemokines and TM4SF, indicating a hybrid nature. CMTM proteins exert an impact on tumor growth by attracting and activating immune cells (*Wang et al., 2022b*). Furthermore, CMTM proteins are capable of regulating the cell cycle, influencing the EGFR-associated pathway and the EMT process, subsequently affecting the proliferation, invasion, and metastasis of tumor cells, indicating their crucial role in tumorigenesis and potential as therapeutic targets (*Wu et al., 2020*). In addition to participating in tumorigenesis, the CMTM family also plays crucial roles in other hematopoietic, immune, cardiovascular, and male reproductive systems conditions (*Chrifi et al., 2017*; *Li et al., 2010*, *2006*; *Zhang et al., 2017a*, *2016a*). Several members of the CMTM family are either overexpressed or inhibited in various tumors, such as pancreatic and gastric cancer, which affect cell proliferation and patient survival, suggesting their involvement in tumorigenesis and prognostic value (*Liang et al., 2021*; *Zhou et al., 2021*). Gene methylation is the

**Table 1 Characteristics of the CMTM family.**

| CMTM | Location | Mechanism | Main subcellular locations | Clinical significance | Reference |
|---|---|---|---|---|---|
| CMTM1 | 16q22.1 | It causes tumorigenesis by affecting cell proliferation. | Plasma membrane, peroxisome, nucleus, extracellular space | It is associated with chemo resistance and poor prognosis of patients. | *Si et al. (2017)*, *Wang et al. (2014)* |
| CMTM2 | 16q22.1 | It affects HIV-1 transcription through AP-1 and CREB pathways. | Plasma membrane, peroxisome, nucleus, extracellular space, Golgi apparatus, cytosol | It provides a new way to control HIV-1 transcription. | *Zhang et al. (2016b, 2017b)* |
| CMTM3 | 16q22.1 | Allele inactivation or methylation makes it lose its ability to negatively regulate cell proliferation. | Plasma membrane, nucleus, extracellular space, endosome | It may be an independent prognostic indicator for gastric cancer. | *Lu et al. (2018)*, *Yuan et al. (2016)* |
| CMTM4 | 16q22.1 | It regulates the cell cycle and affects tumor cell proliferation through synergistic protection of PD-L1. | Plasma membrane, nucleus, extracellular space, Golgi apparatus | It is an effective target for clear cell renal cell carcinomas (ccRCCs) treatment and a factor of poor prognosis. | *Bei et al. (2017)*, *Mezzadra et al. (2017)* |
| CMTM5 | 14q11.2 | It is involved in certain signaling pathways related to tumorigenesis and development. | Plasma membrane, extracellular space | It may be a new target for tumor gene therapy. | *Cai et al. (2017)*, *Guan et al. (2018a)*, *Zhang et al. (2017a)* |
| CMTM6 | 3p22.3 | It cooperates with PD-L1 to participate in immune escape. | Plasma membrane, extracellular space, cytoskeleton, cytosol, lysosome, endosome | It may be a potential immunotherapy target, such as ovarian cancer. | *Burr et al. (2017)*, *Yin et al. (2022)* |
| CMTM7 | 3p22.3 | Rab5 controls the EGFR-AKT signaling pathway and affects tumor development. | Plasma membrane, extracellular space | It is related to the occurrence and development of gastric cancer. | *Jin, Qin & Jia (2018)*, *Liu et al. (2015)* |
| CMTM8 | 3p22.3 | It influences the EGFR signaling pathway through MARVEL to regulate cell proliferation, differentiation and apoptosis. | Plasma membrane, nucleus and extracellular space | It is involved in various tumors and is a new target for tumor gene therapy. | *Both et al. (2014)*, *Gao et al. (2015)* |

underlying mechanism of CMTM5, as promoter methylation could silence CMTM5, and the restoration of CMTM5 overexpression can activate the apoptosis pathway to induce tumor cell death (*Guo et al., 2009*; *Zhou et al., 2021*).

## CMTM1

CMTM1 is located on q22.1 of human chromosome 16, and cDNA sequencing has revealed at least 23 selectively spliced subtypes, among which CMTM1_V17 is the most expressed form (*Siegel et al., 2023*). The upregulation of CMTM1_V17 can confer chemoresistance to non-small cell lung cancer (NSCLC) patients who receive neoadjuvant chemotherapy, while high expression of this subtype in NSCLC tissues is associated with significantly poor prognosis in NSCLC patients (*Si et al., 2017*). Furthermore, overexpression of CMTM1 in the glioma cell line A172 promotes tumor cell proliferation and migration, possibly through activation of epidermal growth factor receptor (EGFR), SRC kinase, and WNT signaling pathways (*Wu et al., 2019*).

## CMTM2

CMTM2 is located on q22.1 of human chromosome 16. It is widely expressed in normal tissues, with a high level of expression in testicular tissues, followed by pancreatic tissue, bone marrow and other tissues (*Kang et al., 2019*). CMTM2 is located in the endoplasmic reticulum near the Golgi apparatus. The MARVEL structure of CMTM2 is related to its function as a carrier protein, as well as its role in cell membranes, which are related to the transport of intracellular substances during steroid synthesis. It also plays an important role in the production of testosterone (*Wu et al., 2020*). CMTM2 can inhibit HIV-1 transcription to some extent by targeting the AP-1 and CREB pathways, as well as affect sperm and testosterone production (*Guo et al., 2020*).

## CMTM3

CMTM3, located on human chromosome 16 q22.1, has a leucine zipper structure and is widely expressed in normal tissues. It participates in cell proliferation, differentiation and development. It can also inhibit excessive cell proliferation, regulate cell migration and suppress tumorigenesis (*Li, Yu & Feng, 2023*; *Zhang et al., 2023b*). CMTM3 is often downregulated or completely silenced in tumor cells, which is closely related to CpG islands and methylation of its promoter (*Ogawa et al., 2012*; *Wang et al., 2009*). CMTM3 can promote the degradation of EGFR to reduce its expression, activate caspase-3 to induce cell apoptosis, and partially inhibit the JAK2/STAT3 signaling pathway to suppress tumor cell proliferation. Thus, it is a potential target for tumor therapy (*Li & Zhang, 2017*; *Lu et al., 2018*; *Wang et al., 2009*; *Yuan et al., 2016*).

## CMTM4

CMTM4, also located on chromosome 16 q22.1, is the most conserved chemokine with three selectively spliced subtypes. It can induce cell accumulation at the G2/M phase and inhibit the proliferation of HeLa cells without inducing apoptosis, revealing its important role in cell proliferation and cell cycle regulation (*Chrifi et al., 2019*; *Xue et al., 2019*). CMTM4 has a synergistic and protective relationship with PD-L1 in tumor tissues. CMTM4 can effectively protect programmed death-ligand 1 (PD-L1) as a target for lysosomal degradation and prevent the clearance of tumor cells by immune cells, suggesting that CMTM4 plays an important role in tumor immunotherapy (*Chui et al., 2022*; *Mezzadra et al., 2017*; *Zhang et al., 2022a*).

## CMTM5

In contrast to the other family members, CMTM5 is independently located on human chromosome 14 q11.2, and there are at least six selectively spliced subtypes, with CMTM5_V1 as the main expressed form (*Li et al., 2020b*). Methylation-mediated promoter silencing can inhibit the expression of CMTM5, but demethylation drugs can restore its expression in tumor cells, thereby effectively inhibiting the cloning, proliferation, adhesion, migration and invasion of tumor cells. CMTM5 overexpression has a significant inhibitory effect on the formation of tumor cell line colonies (*Li et al., 2022a*; *Yuan et al., 2020*; *Zhang et al., 2017a*). Furthermore, CMTM5 induces apoptosis of

pancreatic cancer cells by activating caspases 3, 8, and 9 (*Wu, 2020*). Thus, CMTM5 is involved in several signaling pathways related to tumorigenesis and may be a potential target for tumor therapy (*Ma et al., 2019*; *Yuan et al., 2012*).

## CMTM6

CMTM6, a noncharacteristic protein on the cell surface, is located in a region rich in tumor suppressor genes. CMTM6 acts as a key regulator of PD-L1 in various tumor cells, binding to PD-L1 and maintaining its expression on the cell surface. CMTM6 can prevent PD-L1 from becoming a target of lysosomal-mediated degradation. In addition, CMTM6 participates in immune function by regulating T lymphocyte-mediated antitumor immunity, which indicates that CMTM6, as a specific therapeutic target, can enhance antitumor immunity to a certain extent (*Wang et al., 2022a*; *Zhu et al., 2019*).

## CMTM7

CMTM7 is widely expressed in various tissues but is downregulated or not expressed in some malignant tumors, such as pancreatic and esophageal cancer, suggesting that CMTM7 may participate in tumor development and metastasis, play a tumor suppressive role, and behave as an inhibitory site for many tumors (*Chen et al., 2023*; *Xie, Cheng & Zhang, 2023*). SOX10 is a transcriptional regulator of CMTM7, which is weakly expressed in breast and pancreatic cancer, where it behaves as a tumor suppressor gene (*Jin, Qin & Jia, 2018*; *Li et al., 2014a*; *Lu et al., 2021*; *Zhou et al., 2021*). There are typical CpG islands in the promoter region of CMTM7. Promoter methylation is the mechanism by which CMTM7 expression is downregulated, and CMTM7 expression can affect cell proliferation and tumor development (*Jin, Qin & Jia, 2018*). CMTM7 can reduce AKT phosphorylation and suppress ERK activation, thereby inhibiting PI3K/AKT downstream targets in KYSE180 cells (*Wu, 2020*). Further analysis has shown that CMTM7 can inhibit AKT signaling and induce cell cycle arrest (*Huang et al., 2019*). It can also inhibit the G1/S phase transition by upregulating P27 and downregulating CDK2 and CDK6 (*Li et al., 2014a*).

## CMTM8

CMTM8 is a novel chemotactic cytokine composed of 173 amino acids (*Ge, Duan & Deng, 2021*). It is notable that MARVEL transmembrane domain protein expression is required for CMTM8 to enhance the ligand-induced clearance of EGFR from the cell surface, thereby reducing ERK phosphorylation and affecting the EGFR-mediated signaling pathway (*H'ng et al., 2020*). CMTM8 overexpression causes a change in mitochondrial caspase expression, ultimately inducing tumor cell apoptosis. Furthermore, CMTM8 enhances the migration and invasion of pancreatic cancer cells, possibly through β-catenin signaling activation in pancreatic cancer cells, which in turn promotes pancreatic cancer invasiveness. Notably, reducing CMTM8 expression can lead to the inhibition of pancreatic cancer metastasis, thus establishing CMTM8 as a potential therapeutic target for pancreatic cancer (*Gardner et al., 2013*; *Shi et al., 2021*).
# ASSOCIATION OF THE CMTM FAMILY WITH SEVERAL CANCERS

CMTM proteins are associated with several cancers, such as pancreatic, breast, gastric, liver cancer, and head and neck squamous cell carcinoma (Fig. 1).

## CMTM proteins in pancreatic cancer

Pancreatic cancer is a type of gastrointestinal malignancy that mainly originates from the pancreatic ductal epithelium and acinar cells. It can be categorized as either endocrine cancer or exocrine cancer. This disease is characterized by insidious occurrence, rapid progression, poor therapeutic effects, and poor prognosis (*Chen et al., 2021*; *Ilic & Ilic, 2016*; *Jin et al., 2019*). The mortality rate of pancreatic cancer patients ranks sixth among all deaths caused by malignant tumors in China (*Siegel, Miller & Jemal, 2019*). Pancreatic cancer, whose prevalence is increasing, is expected to overtake breast, prostate, and colon cancer by 2030 to become the second leading cause of cancer death worldwide, only second to lung cancer (*Li et al., 2019*). The occurrence and development of pancreatic cancer is accompanied by a large number of gene mutations, including K-ras, TP53, SMAD4 and CDKN2A, and more than 90% of pancreatic intraepithelial neoplasia cases show K-ras mutations and EGFR overexpression, while approximately 10% of cases are inherited (*Gu et al., 2020*; *Khan et al., 2021*; *Liu et al., 2023*). Pancreatic cancer has a low early diagnostic rate and a low responsiveness to drugs. In recent years, with the in-depth study of tumor immunity and tumor therapy, immunotherapy has become a potential treatment modality after surgery, radiotherapy and chemotherapy (*Schizas et al., 2020*).

Studies on the role of the CMTM family in pancreatic cancer are limited, but some preliminary explorations have been made. CMTM4 can negatively regulate PAK4 to inhibit the PI3K/AKT pathway, which in turn inhibits cell proliferation, and high CMTM4 expression can inhibit pancreatic cancer (*Li et al., 2020a*). Ectopic expression of CMTM8 enhances the migration and invasive ability of pancreatic cancer cells (*Shi et al., 2021*). CMTM5 and TNF-α synergistically induce apoptosis in pancreatic cancer cells (*Wu, 2020*). Metastatic pancreatic cancer is likely to spread to the liver, whose cells express CTLA-4, programmed death receptor-1 (PD-1), and PD-L1, all of which inhibit T-cell functions and result in limited treatment options. Immune checkpoint inhibitors can be used to reactivate the body's anti-tumor immune response, while chemokines, such as cytokines, play important roles in tumorigenesis and autoimmune diseases. CMTM1–8 participate in the occurrence and development of the disease, suggesting that new insights remain to be discovered and applied to pancreatic cancer immunotherapy (*Delic et al., 2015*; *Yuan et al., 2017*). CMTM6 is also involved in preventing PD-L1 lysosomal degradation. LTX-315 is an oncolytic peptide that exhibits PD-L1 inhibition-induced anti-pancreatic cancer immunity effect *via* its potential target ATP11B. CMTM6 mediates the interaction between ATP11B and PD-L1 (*Tang et al., 2022*).

## CMTM proteins in gastric cancer

Gastric cancer is a major global disease, with more than one million new cases every year. It is the fifth most diagnosed malignancy in the world. Due to the lack of biomarkers for

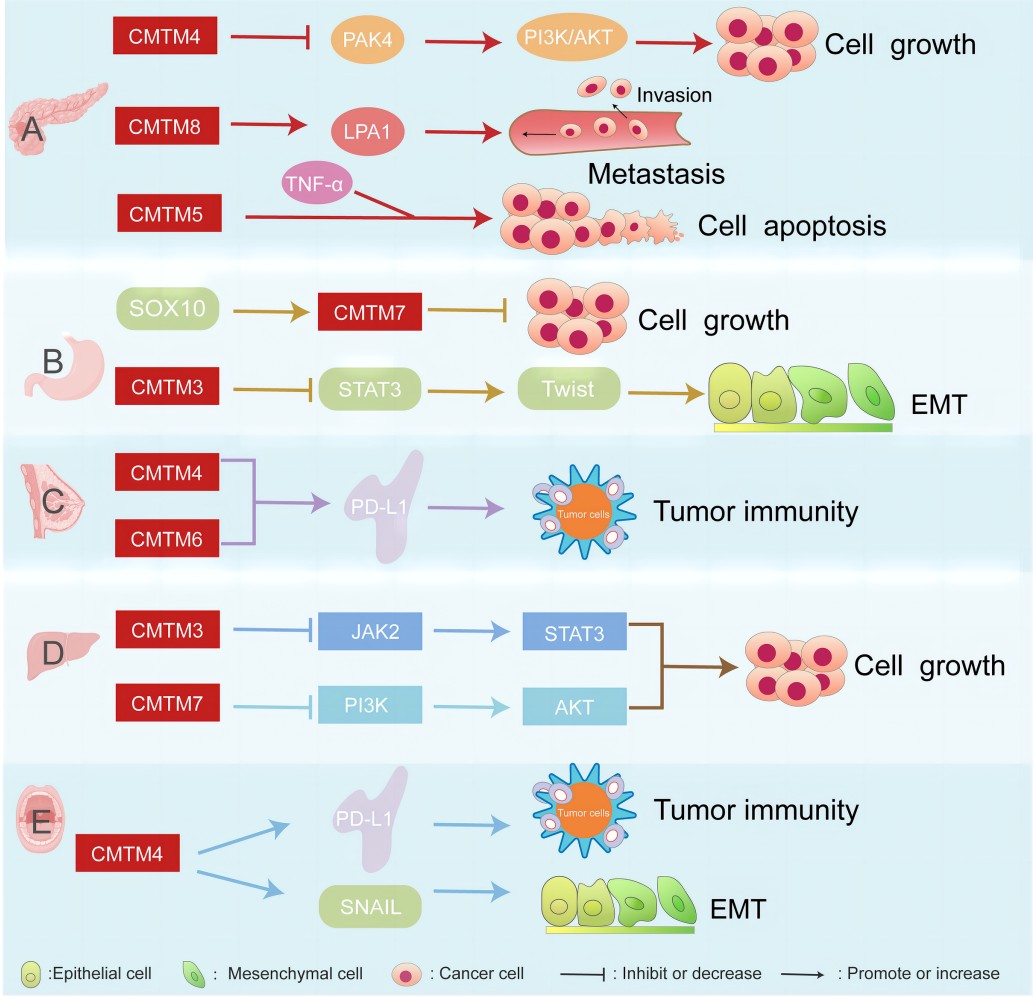

**Figure 1 CMTM proteins have different regulatory network in various cancers.** The CMTM family is associated with pancreatic, breast, gastric, and liver cancer. (A) CMTM4 inhibits PI3K/AKT pathway by negatively regulating PAK4, thus inhibiting the proliferation of pancreatic cancer cells. CMTM8 as an LPA1-related chaperone mediates lysophosphatidic acid-induced metastasis of pancreatic cancer. CMTM5 mediates TNF-α to regulate pancreatic cancer cell apoptosis. (B) SOX10 can regulate the proliferation of gastric cancer by regulating the expression of CMTM7. CMTM3 can inhibit gastric cancer metastasis by regulating the STAT3/Twist1/EMT signaling pathway. (C) CMTM6 affects tumor immunity by regulating PD-L1 on the surface of breast cancer. (D) CMTM3 and CMTM7 inhibit the proliferation of hepatocellular carcinoma cells by inhibiting the JAK2/STAT3 and PI3K/AKT pathways, respectively. (E) CMTM4 is one of the regulators of PD-L1, which is involved in tumour immune regulation, and it is also involved in the EMT process, affecting the expression of some key molecules such as SNAIL.

early diagnosis, most patients are diagnosed at an advanced stage of the disease; therefore, it has a high mortality rate, making it the third most common cause of cancer-related death (*Ajani et al., 2022*; *Bray et al., 2018*; *Smyth et al., 2020*). Approximately 10% of gastric cancer cases have familial aggregation, and 1–3% of gastric cancer patients have germline mutations (*Zhang et al., 2023a*). Studies have shown that CMTM expression is closely related to the occurrence of gastric cancer. According to the analysis obtained by bioinformatics methods, CMTM1, 3, 6, 7, and 8 mRNA levels were up-regulated in GC

tissues, whereas CMTM2, 4, and 5 did not differ from normal tissues (*Liang et al., 2021*). CMTM1 consists of 23 isoforms, of which the CMTM1-v17 protein is up-regulated in expression in various cancers. Although there are no studies related to its association with GC, analysis of clinical data shows that CMTM1 levels correlate with various clinical parameters. Thus, CMTM1 can be used as a biomarker for GC.

A significant mutation in CMTM2 was identified after whole exome sequencing of 23 cases of diffuse-type gastric cancer and non-cancerous paired tissues, and CMTM2 expression predicted prognostic outcome in diffuse-type gastric cancer but not intestinal-type GC (*Choi et al., 2018*). CMTM3 expression is silenced or downregulated in gastric cell lines and primary tumors; it could inhibit EMT by upregulating E-calcineurin expression and downregulating N-calcineurin, waveform protein, and Twist1 expression, suggesting that CMTM3 inhibits gastric cancer metastasis by regulating the STAT3/Twist1/EMT signaling pathway (*Huang et al., 2019*; *Yuan et al., 2016*). CMTM2 and CMTM3 have also been reported to mediate the effects of non-coding RNAs on GC. CMTM2 is involved in the LINC01391/miR-12116/CMTM2 axis and inhibits GC aerobic glycolysis and tumorigenesis (*Qian et al., 2020*). CMTM3, on the other hand, mediates the promoting effects of miR-135b-5p on GC proliferation (*Lu et al., 2018*). The predominant variant of CMTM5, CMTM5-v1, is widely expressed in average human adult and fetal tissues but is undetectable or down-regulated in most cancer cell lines. Like CMTM3, promoter methylation was detected in almost all silenced or down-regulated cell lines (*Ge, Duan & Deng, 2021*). CMTM5 has tumor suppressor activity but is frequently inactivated by promoter methylation (*Shao et al., 2007*).

Recently, studies have found that CMTM6 and PD-L1 expression correlate and are associated with poor prognosis and immune tolerance in GC (*Nishi et al., 2021*). The co-detection of CMTM6 and PD-L1 can be a prognostic indicator for patients diagnosed with GC (*Zhang, Zhao & Wang, 2021*). CMTM6 and CMTM4 are critical regulators of PD-L1 that protect it from lysosomal explication. Thus, they have the potential to serve as indicators to guide the administration of PD-1/PD-L1 therapies or to be used in pre-immunotherapy screening (*Wang et al., 2021*). CMTM7, widely expressed in normal gastric tissues, is weakly expressed in gastric cancer tissues (*Abadi et al., 2021*; *Li et al., 2014a*). SOX10, a transcriptional regulator of CMTM7, mediates CMTM7 expression in gastric cancer. SOX10 overexpression in cell lines with silenced CMTM7 expression can significantly inhibit cell proliferation and gastric cancer progression. Thus, SOX10 can regulate cell proliferation and gastric cancer progression by regulating the expression of CMTM7 (*Jin, Qin & Jia, 2018*). CMTM8 protein is down-regulated in GC tissues, correlates with GC metastasis and patient prognosis, and is an independent protective factor for overall survival (*Yan et al., 2021*).

## CMTM proteins in breast cancer

Currently, breast cancer is the most frequently diagnosed cancer, and the sixth leading cause of cancer-related death among Chinese women (*Richman et al., 2017*). The causes are related to reproductive and hormonal factors such as long menstrual lifespan (early-onset menstruation and later-onset menopause), nutrition, advanced maternal age and

limited breastfeeding (*Bayraktar & Arun, 2019*; *Builder, 2021*; *Moore, 2022*; *Yi, 2021*). The prognostic challenge with breast cancer occurs when patients wait too long before starting treatment, especially if the delay leads to stage and disease progression or leads to more treatment complications. A 2013 study (*Smith, Ziogas & Anton-Culver, 2013*) reported that wait times of more than 6 weeks for starting surgical treatment resulted in a 5-year survival of 80%, while wait times of less than 2 weeks resulted in a 5-year survival of 90% (*Fan et al., 2014*). The CMTM family is associated with the occurrence and development of breast cancer. CMTM5 and seven are biomarkers and prognostic factors of breast cancer. CMTM1_V17 is the most studied CMTM family member in breast cancer, and it is highly expressed in a variety of tumors, including breast cancer (*Kanathezath et al., 2021*). CMTM4, as a PD-L1 protein regulator, can also be a potential surgical target for breast cancer (*Wang et al., 2022b*). In breast cancer patients, the expression level of CMTM5 and CMTM7 is decreased, while the expression level of CMTM6 is increased. Thus, high expression of CMTM7 may have a better prognostic effect. Elevated expression of CMTM6 in human epidermal growth factor receptor 2-positive (HER2+) BC stabilizes the HER2 protein by impeding the ubiquitination process. This heightened stability of the HER2 protein contributes to trastuzumab resistance, thus underscoring CMTM6 as a promising therapeutic target for overcoming resistance (*Xing et al., 2023*). CMTM5 is positively correlated with longer survival time in triple-negative breast cancer (TNBC), a more aggressive type with poor prognosis, which is consistent with the overexpression of the tumor suppressor gene p53 in TNBC patients in earlier studies (*Chen et al., 2020a*; *Dimas-González et al., 2017*). The expression of CMTM7 is higher in TNBC patients, and they show a higher response to chemotherapy and immunotherapy. Additionally, high expression of CMTM7 is positively correlated with immunomodulators, tumor-infiltrating immune cells (TIICs), and immune checkpoint activation (*Jiang et al., 2022*).

Epithelial-mesenchymal translation (EMT) confers aggressive and evasive characteristics to tumor cells. Activation of the EMT process in breast cancer cells can upregulate the expression of CMTM6, which is an essential protein for the cell surface expression of PD-L1. Studies have shown a positive correlation between EMT markers and two other members of the CMTM family, namely, CMTM3 and CMTM7 (*Zhu et al., 2020*). Compared with epithelial cells, CMTM6 and PD-L1 are both overexpressed in mammary mesenchymal cells. The expression level of CMTM1, 2, 4 and 8 in breast cancer are decreased, while the expression level of CMTM5 is not different between healthy individuals and patients with metastatic breast cancer. MiR-182-5p within the extracellular vesicles of BC can specifically target and suppress the expression of CMTM7, activating the CMTM7/EGFR/AKT pathway and facilitating the progression of breast cancer (*Lu et al., 2021*). EMT inhibitors can be used in combination with PD-L1 blockers to improve the response rate (*Xiao et al., 2021*). Modulators of the EMT process, as well as modulators of CMTM6 or CMTM7, can be used in combination with the anti-PD-L1 antibody in patients with highly aggressive breast cancer. In mesenchymal breast cancer cells, CMTM6 silencing decreases PD-L1 expression on the cell surface, while dual targeting of CMTM6 and CMTM7 significantly reduces PD-L1 expression (*Xiao et al., 2021*), suggesting that CMTM-targeted molecular therapy can improve the prognosis of breast cancer patients.

## CMTM proteins in liver cancer

Liver cancer, including hepatocellular carcinoma (HCC) and intrahepatic cholangiocarcinoma, is the sixth most common malignancy and the third most common cause of cancer death worldwide (*Lazzaro & Hartshorn, 2023*; *Ma et al., 2021*), and its incidence is on the rise (*Liu & Liu, 2022*). High rates of invasion and metastasis are the main factors leading to the poor prognosis of HCC patients, and EMT is believed to be the main mechanism underlying cancer cell migration and invasion (*Zhang et al., 2022b*). In HCC, there is a notable increase in the expression of CMTM1, CMTM3, CMTM4, CMTM7, and CMTM8, indicating the pro-tumorigenesis effect of them. Conversely, the expression of CMTM2, CMTM5, and CMTM6 exhibited a significant decrease (*Li et al., 2023a*). *In vivo* study, it has been elucidated that the pro-tumorigenesis effect of CMTM3 is attributed to its ability to upregulate peroxisome proliferator-activated receptor γ (PPARγ) and subsequently activate the PPAR pathway (*Wang et al., 2023*).

CMTM3 inhibits the migration and invasion of HCC cells by inhibiting the EMT phenotype and the JAK2/STAT3 signaling pathway. According to one study, targeted therapy may be an effective modality for the treatment of liver cancer. First, the low expression of CMTM3 in HCC cell lines can significantly inhibit HCC cell proliferation and invasion as well as the EMT process. In addition, CMTM3 overexpression significantly downregulates the phosphorylation levels of JAK2 and STAT3 in HepG2 cells, which suggests that CMTM3 suppresses HCC cell proliferation and tumorigenesis by inhibiting the JAK2/STAT3 signaling pathway. Thus, CMTM3 may be a potential target for the prevention and treatment of HCC (*Li & Zhang, 2017*).

Moreover, immune evasion also contributes to the difficulty of treating liver cancer, and immunotherapy developed to target the tumor immune microenvironment (TIME) kills cancer cells by activating their own immune system. The immune microenvironment plays an important role in the development and treatment of cancer, with tumor-infiltrating T cells, as its key members, recognizing and killing cancer cells (*Chi et al., 2023*), and exosomes, which carry a variety of molecules, acting as a bridge between the various cells in the TIME that communicate with each other (*Chen et al., 2022*). A full understanding of the TIME facilitates the development of new therapeutic targets, of which PD-1/PD-L1 is a relatively representative immune detection site, and in liver cancer, CMTM6 can stabilize PD-L1 and enhance the efficacy of immunotherapy (*Liu et al., 2021*). In HCC, CKLF was found to be involved in the infiltration of immune cells and acted as an independent prognostic biomarker (*Li et al., 2023a*). Furthermore, genes in the CMTM family, such as rs3811178 of CMTM5 and rs164207 of CMTM6, may regulate the HCC risk individually or in combination (*Bei et al., 2018*). CMTM4, located near human chromosome 16q22.1, is often deleted in various tumors, including those in the liver and breast, suggesting the presence of a TSG at this site. The role of TSG is closely associated with tumor occurrence and development and can be inactivated through genetic and epigenetic mechanisms, including promoter CpG methylation and histone modification (*Desaulniers et al., 2021*; *Li et al., 2023b*; *Pal, 2022*). CMTM7 expression is significantly decreased in HCC tissues and negatively correlated with TNM staging. Forced expression of CMTM7 can inhibit the

growth and migration of HCC cells by inhibiting AKT signaling and inducing cell cycle arrest at the G0/G1 phase (*Huang et al., 2019*). This may be possibly due to decreased expression of cyclin D1, cyclin-dependent kinase 4 (CDK4) and CDK6 and increased expression of p27, indicating that CMTM7 acts as a tumor suppressor by inhibiting cell cycle progression in HCC.

## CMTM proteins in oral cancers

It is worth mentioning that CMTM3, as an oncogene, is expressed at low levels in cancer, yet its potential promoter has high methylation levels, and aberrant promoter methylation may be an early signal of tumor development. In laryngeal squamous cell carcinoma (LSCC), hypermethylation of the CMTM3 promoter was found to increase the risk of LSCC in men, especially in the smoking population and in older adults over 55 years of age (*Shen et al., 2016*). The role of CMTM4 is difficult to define, as it exhibits different functions in different tumors, for example, it inhibits the growth of pancreatic cancer cells but its expression level is elevated in type I renal clear cell carcinoma. In head and neck squamous cell carcinoma (HNSCC), CMTM4 has a tendency to be overexpressed and unsurprisingly, it can stabilize PD-L1 expression, while CMTM4 also promotes the process of EMT and affects its related molecules. In addition, CMTM4 can regulate the cancer stem cell -like phenotype through the AKT pathway, which would enhance the significance of CMTM4 as a therapeutic target (*Li et al., 2021*).

# FUNCTIONS OF CMTM PROTEINS IN TUMOR GROWTH AND IMMUNITY

## CMTM and its regulation of the cell cycle

The cell cycle comprises the G1, S, G2, and M phases. This complex process is regulated by cyclins and cyclin-dependent kinases (CDKs). Moreover, there are checkpoints at G1/S and G2/M phase transitions; thus, DNA can be replicated accurately. CDKs can target these checkpoints and regulate the cell cycle. Researchers can also control the proliferation of tumor cells by regulating cell cycle-related proteins (Fig. 2A) (*Wenzel & Singh, 2018*). CMTM3 is frequently downregulated or silenced in testicular cancer cell lines and tumor tissues but highly expressed in normal testis tissues. In a human seminoma cell line, upon the re-expression of CMTM3 *via* adenovirus delivery (Ad-CMTM3), the infected cells expressed a high level of p21, which led to arrest of the cell cycle at the G2 phase and inhibition of cell growth and migration (*Li et al., 2014b*). Similarly, *Plate et al. (2010)* revealed that CMTM4 expression led to the accumulation of HeLa cells at the G2/M phase, subsequently inhibiting cell proliferation. In addition, CMTM5 resulted in G0/G1 arrest (*Cai et al., 2017*), and CMTM7 caused G1/S phase arrest by upregulating p27 and downregulating CDK2 and 6 (*Huang et al., 2019*; *Li et al., 2014a*).

## CMTM and tumor immunity

Tumor immunity refers to how the immune system responds to tumor antigens and eliminates tumor cells. The tumor genome drives tumor immunity due to gene mutation and transcriptional aberration-generated T-cell neoepitopes. In addition, memory T cells

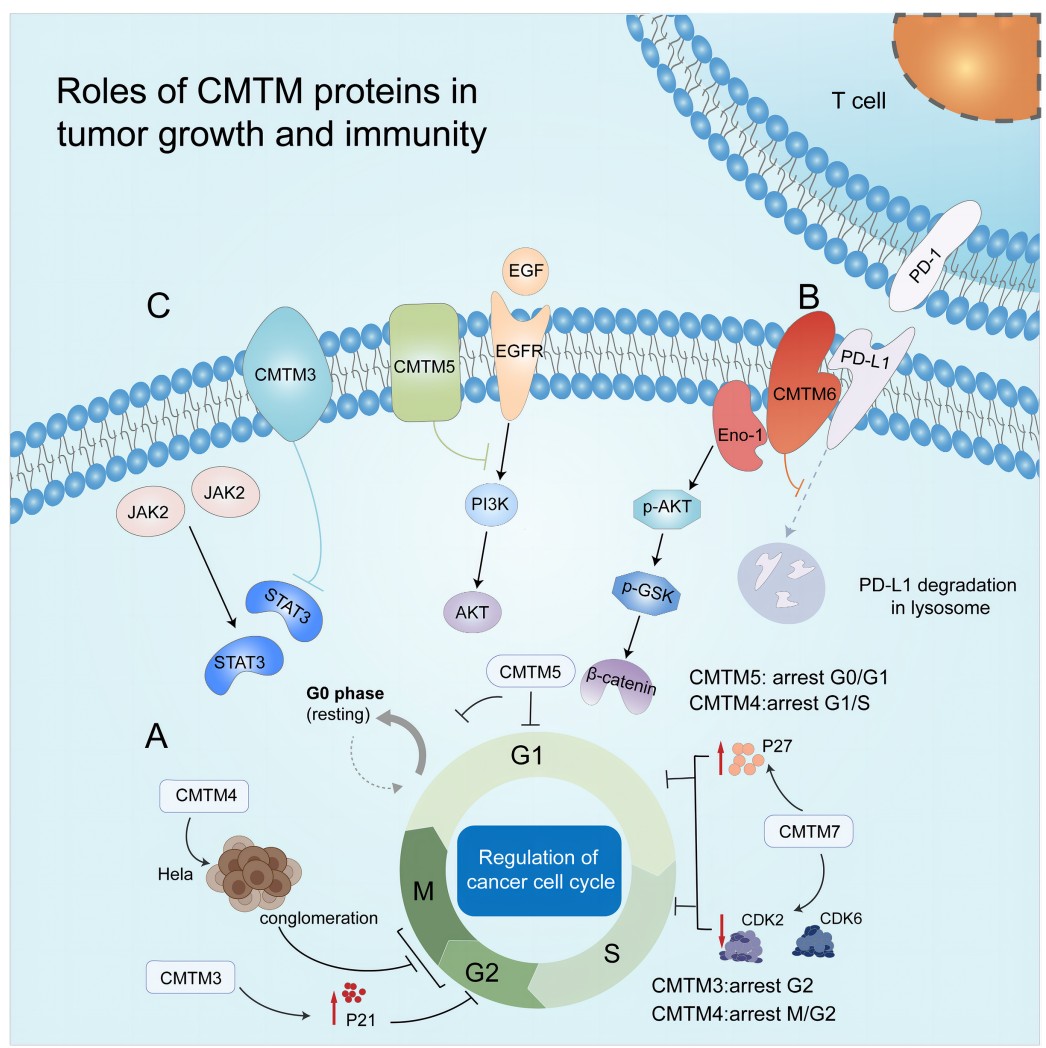

**Figure 2** **Roles of CMTM proteins in tumor growth and immunity.** (A) CMTM protein regulates cell cycle. CMTM3 and four express a high level of p21, leading to cell proliferation inhibition at G2/M phase. CMTM5 results in G0/G1 arrest and CMTM7 causes G1/S phase arrest by upregulating p27. (B) CMTM6 induces immune escape of tumor cells through tissue lysosomal mediated degradation of PD-L1. (C) The CMTM family mediates tumor-related signaling pathways and can be used as targets for cancer therapy.

play a vital positive role in tumor immunity (*Chen & Mellman, 2017*). Of note, tumor cells evolved to express immune checkpoint ligands, thus inhibiting the cytotoxicity of the immune system. As one of the classical ligands, PDL-1 interacts with the checkpoint PD-1 on T cells and inhibits T cell activity (*Kornepati, Vadlamudi & Curiel, 2022*).

Beyond PD-L1 inhibitory therapy, the CMTM family may also have other functions in tumor immunity. For instance, CMTM6 is involved in immune functions by modulating T lymphocyte-mediated antitumor immunity. In a bioinformatics study that conducted gene set variation analysis of transcriptome data from 1,862 glioma samples available from CGGA RNA sequencing, TCGA RNA sequencing, CGGA microarrays, GSE16011 data, and IVY GBM databases, CMTM6 expression was positively correlated with

immunosuppressive factors such as induced T-cell tolerance, cytokine synthesis and secretion, and regulatory T-cell differentiation (*Guan et al., 2018b*). This study also revealed that CMTM6 expression may be positively correlated with inflammatory responses and somatic mutations that promote cancer progression.

Furthermore, CKLF1 expression is upregulated in activated CD4[+] and CD8[+] lymphocytes, indicating that its expression is regulated in various autoimmune diseases (*Duan et al., 2020*). CMTM7 regulates B-cell function in terms of tumor immunity. In CMTM7 knockout mice, decreased expression level of BCR on the B1a cell surface and decreased serum IgM levels were observed compared to normal mice (*Liu et al., 2020*). With the rapid development of immunotherapy, a number of new therapeutic directions have emerged; IL-7 plays an integral role in the growth, development and even maintenance of memory of T cells, demonstrating its potential as an adjuvant in combination with cancer vaccines (*Zhao et al., 2022*). Exosomes have great potential in cancer immunotherapy as a bridge of communication, capable of loading various molecules to deliver information; they can be used as potential markers to aid diagnosis, and targeted drugs can be investigated to target tumor exosomes, thus opening up more possibilities for immunotherapy (*Gong et al., 2022*).

## CMTM AND TUMOR IMMUNOTHERAPY

PD-L1 is expressed on the surface of various cells, including tumor cells and myeloid cells (*Munoz et al., 2021*). Anti-PD-1 therapy aims to dampen the interactions between tumor-reactive T cells and tumor cells by blocking PD-1 ligand/PD-1 signaling (*Topalian, Drake & Pardoll, 2015*). In different tumors, blocking PD1 and PD-L1 interactions has shown great clinical benefits, except for some nonresponsive cases. To promote the clinical efficacy of immune checkpoint therapy, it is necessary to investigate the intrinsic mechanisms related to PD-L1 expression and then to identify potential targets for clinical research (*Topalian et al., 2012*). However, our understanding of the regulation of PD-L1 is limited.

CMTM6 is an uncharacterized protein expressed on the cell surface. To investigate this potentially new regulator of PD-L1 expression, *Burr et al. (2017)* performed a whole-genome CRISPR–Cas9 deletion library screen using the pancreatic cancer cell line BxPC-3 and observed that IFN-γ can indeed stimulate the expression of endogenous PD-L1. Subsequently, the authors demonstrated that CMTM6 colocalizes with PD-L1 on the cell surface with or without IFN-γ stimulation and that this interaction is only maintained under conditions of complete membrane integrity. Notably, among more than 5,000 quantified proteins on the cell surface, CMTM6 is one of four proteins that decreased more than twofold by plasma membrane profiling and other unbiased analytic methods, indicating that CMTM6 is required for PD-L1 expression. Furthermore, after examining PD-L1 maturation in wild-type and CMTM6-knockout cells, they found that CMTM6 maintains stable expression in the plasma membrane but not during translocation from the endoplasmic reticulum to the cell surface. They also demonstrated that a large proportion of surface PD-L1 is continuously internalized and recycled, and CMTM6 plays a role in this recycling by preventing PD-L1 from being targeted for lysosome-mediated

degradation (*Burr et al., 2017*). *Mezzadra et al. (2017)* showed that this function is shared by the closest family member, namely, CMTM4, in HAP1 cells. They also demonstrated that CMTM4 and 6 increased PD-L1 expression without affecting PD-L1 transcript levels but prolonged the half-life of PD-L1.

CMTM6 knockdown leads to decreased expression of PD-L1 in 12 human tumor lines of melanoma, thyroid cancer, colorectal cancer, lung cancer and CML, as well as in three short-term melanoma xenografts (*Mezzadra et al., 2017*). CMTM6 stabilizes PD-L1 expression by preventing lysosome-mediated degradation of PD-L1 (*Mezzadra et al., 2017*). Another study demonstrated that CMTM6 knockdown reduced the expression of PD-L1 and increased the infiltration of CD8$^+$ and CD4$^+$ T cells, which enhanced antitumor immunity in head and neck squamous cell carcinoma (*Chen et al., 2020b*). CMTM6 expression was also upregulated in advanced malignant glioma, and it was responsible for the poor prognosis of patients with glioma (*Guan et al., 2018b*).

In lung cancer patients, CMTM6 expression acts as a predictor of PD-1 inhibitor therapy; that is, patients with higher CMTM6 expression respond well to PD-1 inhibitors (*Gao et al., 2019*; *Koh et al., 2019*). In addition, CMTM6 expression has been detected in several healthy tissues, and it may have functions other than triggering immune evasion by tumor cells (Fig. 2B).

In a retrospective NSCLC cohort (438 individuals, in which 69 were treated with immunotherapy, 258 were untreated, and a collection of EGFR and K-ras genotyped tumors), high coexpression of CMTM6 and PD-L1, particularly in the stroma (*e.g.*, macrophages), predicted the outcomes of PD-1 blockade immunotherapy (*Zugazagoitia et al., 2019*).

In many other cancer types, CMTM6 is an indicator of clinical outcome and prognosis. Recently, a pancancer data analysis of 33 cancer types revealed a close correlation between CMTM6 and PD-L1 in many cancers as well as a correlation with overall survival (*Zhao et al., 2020*). In 185 gastric cancer specimens, CMTM6 and PD-L1 were mainly expressed both on the cell surface and in the nucleus of tumor cells (expression rates were 78.38% and 75.68% respectively), and they may be independent indicators for overall survival (*Zhang, Zhao & Wang, 2021*). In metastatic melanoma, high CMTM6 and PD-L1 coexpression in the stromal compartment was highly correlated with longer survival in treated patients, while PD-L1 expression showed prognostic value in control patients (*Martinez-Morilla et al., 2020*). Likewise, in TNBC, the expression of CMTM6 was correlated with progression-free survival (*Tian et al., 2021*).

Taken collectively, these findings reveal that PD-L1 is dependent on the expression of CMTM4 and six to efficiently carry out its inhibitory function, suggesting that they are potentially new targets to improve the efficacy of immune checkpoint therapy.

## MODULATION OF ANTITUMOR IMMUNITY BY CMTM

CMTM family proteins exert their biological functions by regulating signaling pathways related to cell growth and migration, including the EGFR, WNT, and JAK2/STAT3 signaling pathways, suggesting that CMTM proteins may serve as potential targets for modifying tumor development (Fig. 2C).

## Regulation of the EGFR signaling pathway

EGFR signaling maintains cell homeostasis and regulates epithelial tissue development; therefore, it can be a biomarker of resistance in tumors (*Yarden & Pines, 2012*). EGFR is phosphorylated after stimulation by EGF, triggering the intracellular signaling cascade to control cell proliferation and differentiation. Several CMTMs can regulate this critical signaling pathway (*Sigismund, Avanzato & Lanzetti, 2018*). In gastric cancer, CMTM3 promotes the degradation of EGFR to impede tumor metastasis (*Yuan et al., 2016*), indicating that knocking out the CMTM3 gene can be a potential therapy for gastric cancer. In addition, CMTM3 accelerates the degradation of EGFR, inhibits the EGFR/STAT3/EMT signaling pathway, increases the expression of TP53, and enhances the TP53 signaling pathway in chordoma cells (*Yuan et al., 2021*).

Both CMTM3 and CMTM5 can regulate the endocytic trafficking of EGFR. CMTM3 reduces the expression of EGFR on the cell surface by promoting early endosome fusion by activating Rab5 (*Yuan et al., 2017*). However, in HCC and prostate cancer, CMTM5 inhibits this signaling pathway by targeting its downstream pathway, namely, the PI3K/AKT signaling pathway (*Xiao et al., 2015*). CMTM6, 7 and 8 are widely expressed in human normal adult tissues and normal epithelial cell lines. Only CMTM7 expression is frequently suppressed or reduced in esophageal and nasopharyngeal cell lines, but this is not true for other cancer cell lines. Similar to CMTM3, CMTM7 also regulates the EGFR/AKT pathway in NSCLC (*Liu et al., 2015*). A previous study revealed that CMTM7 can promote the internalization of EGFR and further inhibit the AKT signaling pathway (*Li et al., 2014a*). CMTM8 accelerates the degradation of EGFR from the cell surface by relying on the MARVEL transmembrane domain protein, thereby influencing the EGFR signaling pathway by dampening ERK phosphorylation (*Zhang et al., 2012*).

## Association with WNT signaling

The WNT signaling pathway is vital for embryonic development, tissue regeneration, stem cell homeostasis, and cell proliferation. The WNT pathway is highly conserved throughout evolution, and its downstream effects were traditionally separated into canonical (β-catenin dependent) WNT signaling and non-canonical (β-catenin independent) signaling (*Parsons, Tammela & Dow, 2021*). Alteration of WNT signaling was involved in tumorigenesis of various cancer types, including colorectal cancer, liver cancer, and lung cancer (*Parsons, Tammela & Dow, 2021*).

A previous study demonstrated that the interaction of CMTM6 with membrane-bound enolase-1 stabilized its expression, leading to activation of WNT signaling mediated by AKT–glycogen synthase kinase-3β (*Mohapatra et al., 2021*). These findings indicate that CMTM6 can be a promising target for treating therapy-resistant oral squamous cell carcinoma by facilitating tumor cell immune evasion and reversing cisplatin resistance (*Mohapatra et al., 2021*).

Similarly, the WNT/β-catenin signaling pathway is involved in HNSCC development, including cell proliferation and differentiation with mesenchymal traits as early signs of disease (*Lee et al., 2014*). CMTM6 expression is associated with WNT/β-catenin signaling.

Depletion of CMTM6 affects the maintenance of stemness properties and inhibits TGFβ-induced EMT in HNSCC cells (*Chen et al., 2020b*).

Another study investigated the overexpression of CMTM genes on glioblastoma cell proliferation and invasion and revealed that CMTM1 and 3 promote tumor invasion, and their expression was significantly correlated with shorter overall survival. Consequently, the study showed that CMTM1 and CMTM3 can be targets for treating glioblastoma (*Delic et al., 2015*). Additionally, using a human phosphokinase protein expression profiling assay, the authors demonstrated that CMTM1 and three downstream signaling may be correlated with the expression of growth factor receptor and Src kinases, as well as WNT activation.

## JAK2/STAT3 signaling pathway

The JAK2/STAT3 pathway interacts non-covalently with various cytokine receptors and is activated upon receptors binding to hormones, growth factors, or cytokines. This pathway is pivotal in regulating cellular processes such as proliferation, differentiation, apoptosis, and immune modulation (*Mengie Ayele et al., 2022*). It is implicated in the tumorigenesis and progression of both hematological malignancies and solid tumors due to its involvement in regulating cell growth-related gene expression, exerting adverse effects on prognosis (*Thomas et al., 2015*).

*In vitro*, CMTM3 overexpression inhibited cell proliferation and invasion, and the EMT process in HCC cells. The phosphorylation of JAK2 and STAT3 in HepG2 cells was also decreased. *In vivo*, CMTM3 overexpression inhibited tumor growth in Bal-b/c nude mice. Thus, CMTM3 plays an important role in HCC metastasis by triggering EMT and suppressing the JAK2/STAT3 signaling pathway, indicating that CMTM3 is a potential target in the prevention and treatment of HCC (*Li & Zhang, 2017*).

## Potential of CMTM proteins as therapeutic targets

CMTM1 consists of at least 23 alternatively spliced isoforms designated CMTM1_V1-V23, among which CMTM1_V17 is highly expressed in breast tumors. Moreover, a study revealed that CMTM1_V17 enhances cell proliferation and inhibits TNF-α-induced tumor cell apoptosis (*Wang et al., 2014*). Survival analyses showed that CMTM1 could be an independent *post hoc* factor in hepatocellular carcinoma (*Song et al., 2021*). CMTM1 and CMTM3 are targets for treating glioblastoma (*Delic et al., 2015*). CMTM2 expression is down-regulated and prognostically relevant in HCC and may serve as an independent prognostic factor (*Guo et al., 2020*). Moreover, serum CMTM2 levels can be a valuable indicator of the pathogenesis of HBV-related diseases (*Chen et al., 2020c*). Up-regulation of CMTM2 expression under the effect of Si-Jun-Zi decoction inhibits the tumor properties of gastric cancer (*Li et al., 2022b*). CMTM3 affects EMT progression by inhibiting the EGFR/STAT3 signaling pathway, thereby inhibiting the development of tumorigenesis in chordoma and gastric cancer. CMTM3 could be used as its potential therapeutic target (*Yuan et al., 2021*). Depletion of CMTM4 enhances the sensitivity of HCC tumors to anti-PD-L1 therapy, allowing more patients to benefit from immunotherapy (*Chui et al., 2022*). A previous study reported that low expression of

CMTM5 in hepatocellular carcinoma significantly correlated with poor overall survival (*Xu & Dang, 2017*). Another study suggested the prognostic value of CMTM5 expression and that it might prolong the survival of patients with TNBC (*Chen et al., 2020a*). CMTM6 has been associated with the development of a variety of cancers. In hepatocellular carcinoma, CMTM6 interacts with Vimentin, thereby promoting invasive metastasis of HCC (*Huang et al., 2021*); in breast cancer, CMTM6 stabilizes the HER2 protein and contributes to trastuzumab resistance (*Xing et al., 2023*); and in gastric cancer, CMTM6 stabilizes the expression of PD-L1, and its high expression correlates with poor prognosis of gastric cancer, and the combined assay can be used as a prognostic indicator in gastric cancer (*Zhang, Zhao & Wang, 2021*). CMTM7 can interact with Catenin Alpha 1 to regulate Wnt/β-catenin signaling to inhibit breast cancer progression and be a novel target for breast cancer (*Chen et al., 2023*). CMTM8 can interact with LPA1, activating oncogenic β-catenin signaling as a potential therapeutic target for pancreatic cancer (*Shi et al., 2021*).

## CONCLUSIONS

CMTM family proteins perform critical functions in many biological processes during tumor development and metastasis, including triggering proliferation, resisting cell death, and activating invasion and metastasis. These characteristics all belong to "Hallmarks of cancer: the next generation" proposed by Douglas Hanahan and Robert A. Weinberg in 2011 (*Hanahan & Weinberg, 2011*). Immunotherapy is a relatively new therapeutic modality that has emerged in the last decade or so in addition to conventional surgical resection, radiotherapy and chemotherapy, and in particular, targeting PD-1/PD-L1 has achieved remarkable results in clinical trials in many tumors. CMTM family proteins are involved in the regulation of immune cells and key molecules such as PD-L1, which can be potential targets for immunotherapy.

In this review, we briefly summarize recent findings on the roles of CMTM proteins in tumor growth and their mechanisms, as they relate to the cell cycle, tumor immunity, signaling pathways and their potential as therapeutic targets. CMTM3, 4, 5 and 7 play important roles in the cell cycle, leading to cell cycle arrest, tumor growth and migration inhibition. Blocking PD-1/PD-L1 *via* monoclonal antibodies has shown outstanding clinical efficacy in patients with various tumors; however, there are still some nonresponsive individuals with poor outcomes. Notwithstanding the significance of PD-L1 expression by cells within the tumor microenvironment, our understanding of the regulation of the PD-L1 protein is limited. Current studies have focused on the mechanisms of action of CMTM proteins as PD-L1 protein regulators. Multiple studies have reported that PD-L1 is dependent on CMTM4 and 6 to efficiently carry out its inhibitory function, suggesting that they are potentially new targets to improve the efficacy of immune checkpoint therapy. CMTM family proteins also exert their biological functions by signaling pathways that correlate with cell growth and migration, including the EGFR, WNT and JAK2/STAT3 signaling pathways, indicating that CMTM proteins are potential targets for altering tumor progression and metastasis. Although we have consolidated a large amount of research to argue our point, our study does have some limitations that need to be addressed in the future. First, the current research on the

CMTM family is inherently limited; the role of the CMTM family in tumor development has been validated, but the exact mechanisms remain unclear, and there are no robust experimental data to validate this claim. More studies are needed in the future to further explore the expression of CMTM in tumors, the related molecular mechanisms and signaling pathways and to provide new strategies for tumor immunotherapy.

Recently, some studies have focused on the CMTM family and its role in cancer biological processes (*Lu et al., 2016*; *Wu et al., 2020*; *Xie, Cheng & Zhang, 2023*). However, this review offers unique advantages compared to existing research. Firstly, this review provides a more comprehensive analysis by examining various different types of cancers, including most of the abdominal, thoracic and oral tumors, whereas the review by *Xie, Cheng & Zhang (2023)* focuses more on NSCLC. Furthermore, this review individually introduces each member of the CMTM family, followed by a detailed description of their specific mechanisms in specific cancers, rather than providing broad generalizations, and finally highlight the therapeutic potential of CMTM family. Additionally, considering the expanding clinical applications of tumor immunology, our review offers a more comprehensive understanding of the correlation between CMTM family and immunity, encompassing various mechanisms, including PD-L1/PD-1, EGFR, WNT, and JAK2/STAT3 pathways. Moreover, our review provides updated information and insights into the molecular pathways through which CMTM influences different types of cancer to ensure innovation.

As described in this article, CMTM proteins play an important regulatory role in tumor development as well as in tumor immunity and have the potential to become novel targets for immunotherapy. In addition, the expression of CMTM proteins can also be used to characterize the immunological environment of patients, especially CMTM6, which is closely related to PD-L1, to appropriately estimate patient prognosis and help physicians identify subgroups of patients sensitive to immunotherapy for individualized treatment.

## ACKNOWLEDGEMENTS

We thank all authors for their contributions to this review.

### Funding
The authors received no funding for this work.

### Competing Interests
The authors declare that they have no competing interests.

### Author Contributions
- Sai-Li Duan conceived and designed the experiments, prepared figures and/or tables, authored or reviewed drafts of the article, and approved the final draft.
- Yingke Jiang conceived and designed the experiments, prepared figures and/or tables, authored or reviewed drafts of the article, and approved the final draft.

- Guo-Qing Li performed the experiments, prepared figures and/or tables, authored or reviewed drafts of the article, and approved the final draft.
- Weijie Fu performed the experiments, prepared figures and/or tables, and approved the final draft.
- Zewen Song analyzed the data, authored or reviewed drafts of the article, and approved the final draft.
- Li-Nan Li analyzed the data, authored or reviewed drafts of the article, and approved the final draft.
- Jia Li analyzed the data, authored or reviewed drafts of the article, and approved the final draft.

## Data Availability

This article is a literature review.

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
