# Peer review of "Research insights into the chemokine-like factor (CKLF)-like MARVEL transmembrane domain-containing family (CMTM): their roles in various tumors"

_PeerJ, doi:10.7717/peerj.16757_

## Round 0.1 · original submission · Major Revisions

Dear Author, we received the reviewers' reports, and on the basis of the reports I recommend major revision. Kindly address the reviewers' queries and make the rebuttals.

Reviewer 1 ·

Basic reporting

Please see Additional Comments.

Experimental design

Please see Additional Comments.

Validity of the findings

Please see Additional Comments.

Additional comments

The authors reviewed the role of CMTM family members in various tumors, specifically in the context of tumor immunotherapies. Overall, this study is suitable for publication, only if the authors address the following issues:

1. Throughout the manuscript, it seems better to use Grammarly (https://www.grammarly.com/) to check & correct potential grammatical errors or typos. For example,
1.1 In ABSTRACT, it seems better to change "In this review, we discuss each CMTM family member according to its location on a specific chromosome; the signaling pathway each member controls or participates in, as well as its upregulation or downregulation; summarize the roles of different CMTM family members in pancreatic, breast, gastric and liver cancers; and discuss the mechanisms of action of the different CMTM family members based on recent research" into "In this review, we discuss each CMTM family member's chromosomal location, involved signaling pathways, expression patterns, and potential roles, and mechanisms of action in pancreatic, breast, gastric and liver cancers", which would be more concise and easier to read.

2. To help readers to get a better understanding of the current review's novelty — how this review advances our previous knowledge about the CMTM family in tumors, it would be more informative to compare this manuscript with similar reviews (PMID: 32944388, PMID: 27356683, and PMID: 36792080) by summarizing the current review's strengths over others.

3. In ABSTRACT:
3.1 It would be more readable and informative to mention the full names of abbreviations: "CMTM", "CKLF", and "TM4SF", when these acronyms appear in the manuscript for the first time.
3.2 It would be clearer and more informative to rewrite "as well as in the hematopoietic, immune, cardiovascular and male reproductive systems" by specifying the role of CMTM family in these systems — whether CMTM family members play a role in these systems' tumors, other diseases, homeostasis, or development.
3.3 It seems better to change "CMTM family members use a similar mechanism to activate and chemoattract immune cells to affect the proliferation and invasion of tumor cells due to their common structural characteristics with typical chemokines and TM4SF" into "CMTM family members activate and chemoattract immune cells to affect the proliferation and invasion of tumor cells through a similar mechanism, the structural characteristics typical of chemokines and TM4SF", if the revision does not change the meaning that the authors originally wanted to convey.
3.4 It seems better to change "Furthermore, we discuss several targeted therapies that have been clinically applied in recent years" into "Furthermore, we discuss several clinically applied tumor therapies targeted at the CMTM family", which would be clearer and more cohesive (that is, more closely connected to the sentences before & after it).
3.5 It seems better to change "indicating that CMTM family members can be novel immune checkpoints and potential targets" into "indicating that CMTM family members could be novel immune checkpoints and potential targets effective in tumor treatment", which would be more rigorous and cohesive.

4. In INTRODUCTION:
4.1 In "CMTM1–8, which are related to the chemokine family and the transmembrane 4 superfamily (TM4SF)", it would be clearer and more informative to expand on the relationship between CMTM1–8 and "the chemokine family and the transmembrane 4 superfamily (TM4SF)", rather than just saying "are related to".
4.2 In "which impact tumor cell proliferation and invasion and participate in the regulation of cell growth and development", it could be more concise to rewrite or refine this sentence, because "the regulation of cell growth" seems to repeat "impact tumor cell proliferation" — they appear to have a similar meaning. In addition, cell "development" tends to have a different meaning from "cell growth"; if the CMTM family was not been reported to participate in the cell fate decision (a concept in stem cell biology or developmental biology), it would be more accurate to delete the word "development".
4.3 In "the mechanism of action is related to gene methylation, as CMTM overexpression can activate the apoptosis pathway, which leads to induce tumor cell death", please rewrite this sentence to make it more informative and rigorous by replacing the conjunction "as" with another cohesive device that showing a different logical relationship. Because the "gene methylation" did not seem to be supported by the evidence that the authors selected ("CMTM overexpression can activate the apoptosis pathway, which leads to induce tumor cell death").
4.4 In "This review discusses the relationship between CMTM family members and various cancers and discusses the roles of CMTM1–8 and their potential mechanisms of action", it seems better to change this sentence into "This review discusses the roles of CMTM family members in various cancers and the underlying mechanisms of action", which would be more concise and easier to read.
4.5 In "Finally, we discuss the potential of the CMTM family members as therapeutic targets for tumors and their ability to improve patient outcomes", it seems better to change this sentence into "Finally, we discuss the potential of the CMTM family members as therapeutic targets for tumors to improve patient outcomes", which would be more concise.
4.6 In "While different members of the CMTM family have different functions in the pathogenesis of malignant tumors, they also serve important roles in the immune and reproductive systems (Wu et al. 2019a). The proteins encoded by the CMTM1–8 genes possess structural characteristics that are commonly found in typical chemokines and TM4SF", it seems better to change this sentence into "The proteins encoded by the CMTM1–8 genes possess structural characteristics that are commonly found in typical chemokines and TM4SF. CMTM family members have important roles in not only tumor pathogenesis but also reproductive and immune systems (Wu et al. 2019a)", which would be more cohesive and concise.

5. In SURVEY METHODOLOGY:
5.1 In "indicating that CMTM family members can become new immune checkpoints and potential targets", it would be more concise to delete this sentence, which seems irrelevant to the "survey methodology".
5.2 In "We also considered our previous published literature on CMTM cross-references to identify other appropriate relevant resources", it would be clearer to explain how considering "our previous published literature on CMTM cross-references" would help "identify other appropriate relevant resources". In other words, please expand on what were the "other appropriate relevant resources".

6. In 3 CMTM FAMILY:
6.1 In "the MARVEL transmembrane domain protein is a member of the CMTM family", it would be clearer to explain what is the relationship between this sentence and the sentences before it (sentences introducing the "nine members" of CMTM family).
6.2 In "The proteins encoded by CMTM genes share common structural characteristics with typical chemokines and TM4SF. CMTM proteins attract and activate immune cells, which can influence tumor cell proliferation and invasion. These proteins also participate in regulating cell growth and development, indicating their vital role in tumorigenesis and potential as therapeutic targets", it seems better to rewrite these sentences to avoid repeating the expressions exactly the same as INTRODUCTION.
6.3 In "In addition to their various biochemical processes, the CMTM family also plays crucial roles in the hematopoietic, immune, cardiovascular and male reproductive systems (Wu et al. 2019a)", it seems better to change this sentence into "In addition to participating in tumorigenesis, the CMTM family also plays crucial roles in other conditions of hematopoietic, immune, cardiovascular, and male reproductive systems", which would be more cohesive and accurate. In addition, it would be more rigorous to replace the reference ("Wu et al. 2019a") with primary sources (the references 16–20 in the paper by Wu et al. 2019a), rather than the paper by Wu et al. 2019a itself (that is a secondary source).
6.4 In "CMTM proteins attract and activate immune cells, which can influence tumor cell proliferation and invasion" and "These proteins also participate in regulating cell growth and development, indicating their vital role in tumorigenesis and potential as therapeutic targets", it would be more informative and rigorous to support these two statements by citing references.
6.5 In "Several members of the CMTM family are inhibited or deleted in various tumors, such as pancreatic and gastric cancer", it would be more cohesive to rewrite this sentence by appending a clause like ", suggesting ..." to pinpoint how this evidence contributes to our understanding of CMTM family roles in cancer.
6.6 In "Gene methylation is the underlying mechanism of CMTM genes, as CMTM overexpression can activate the apoptosis pathway to induce tumor cell death", it would be more accurate to rewrite this sentence, like the comment 4.3.
6.7 In 3.1 CMTM1, it seems better to rewrite "The upregulation of CMTM1_V17 can confer chemoresistance to tumor cells, while low expression of this subtype in tumor tissues after chemotherapy is associated with significantly longer overall survival times in patients" by specifying the "tumor" type. In addition, adding references would be more informative and rigorous.

7. In 4 ASSOCIATION OF THE CMTM FAMILY WITH SEVERAL CANCERS,
7.1 In 4.1 CMTM proteins in pancreatic cancer, it would be concise to delete "Some studies have shown that dysregulation of lncRNAs affects the functions of pancreatic cancer cells, such as proliferation, apoptosis, promotion of metastasis and evasion of tumor suppressors (Huang et al. 2023). In addition, H. Zhi. et al. found that cuprotosis programmed-cell-death-related lncRNAs can be a novel prognostic biomarker and potential therapeutic target for pancreatic cancer patients (Chi et al. 2022)", which were two studies about lncRNAs that did not seem to be linked to the CMTM family by the authors. Alternatively, please explain why the authors mention these two studies and how the studies could be lined to the CMTM family.

8. In 5.2 CMTM AND TUMOR IMMUNITY's beginning, it would be more informative to briefly introduce tumor immunity.

9. In 7.2 ASSOCIATION WITH WNT SIGNALING's beginning, it would be more informative and easier to understand to briefly introduce WNT signaling and its roles in cancer.

10. In 7.3 JAK2/STAT3 SIGNALING PATHWAY's beginning, it would be more informative and easier to understand to briefly introduce JAK2/STAT3 signaling and its roles in cancer.

Reviewer 2 ·

Basic reporting

I enjoyed reading this manuscript describing the CMTM family proteins and their relevance in cancer biology. The authors defined CMTM family members according to their location on specific chromosomes, their signaling pathways, involvement in cell cycle regulation and tumor immunity, as well as their role in different types of cancers, finally tackling their clinical relevance and their potential use in targeted therapies. Overall, the manuscript provides valuable information, is well-researched, comprehensive, and well written.

Nonetheless, I identified a major issue that requires the authors’ attention.
It is unclear what the authors consider their main contribution to the academic literature, as this specific topic has been tackled in recently published reviews, majorly Wu et al, 2020 “CMTM family proteins 1–8: roles in cancer biological processes and potential clinical value” and Li et al, 2022 “CMTM Family and Gastrointestinal Tract Cancers: A Comprehensive Review”, among others.
This review did not seem to answer a different question or provide significant updates to what has already been published.

Experimental design

No comment

Validity of the findings

No comment

Additional comments

1- P1 Introduction line 27-29 “the mechanism of action is related to gene methylation, as CMTM overexertion can activate the apoptosis pathway, which leads to induce tumor cell death (Wu et al. 2022)”. The reference cited does not provide this information. The authors could have meant Wu et al. 2020.
2- P3 lines 63-65 “Gene methylation is the underlying mechanism of CMTM genes, as CMTM over expression can activate the apoptosis pathway to induce tumor cell death (Guo et al. 2009; Zhou et al. 2021)”. The statement is general while the citations specifically mention CMTM3 and CMTM5
3- Table 1. The authors described the different CMTM family members in details and provided a summary table. The table is similar to the one published by Wu et al, 2020. and has the same title. It includes the mechanism of action of each CMTM member as well as the clinical relevance, which is a good addition, but the information could be updated by using more recent citations e.g. CMTM6 can serve as a novel prognostic biomarker in patients with Ovarian cancer (Yin et al, 2022)
4- Section 4, “Association of the CMTM family with several cancers” contains information that is tackled in the review by Li et al, 2022. Not much new information is provided concerning Gastrointestinal Tract Cancers.
5- Section 4 could be the point of originality of this manuscript by providing more updated information and molecular pathways pertaining to the different types of cancers.
6- Section 4.5, “CMTM proteins in other cancers” should be replaced by” CMTM proteins in oral cancers”.
7- Figure 1 is original and could be the main focus of the review, but it lacks important and new pathways that have been omitted e.g. CMTM5 in pancreatic cancer. Authors should expand on that and maybe include prognostic markers for each type of cancer.
8- The signaling pathways also have been described previously and figure 2A is an adaptation of another figure featured in Wu et al, 2020. I suggest relating the relevant molecular pathways to the specific types of cancers discussed previously.
9- Section 7.4 “Potential of CMTM proteins as therapeutic targets” is underdeveloped considering it’s a major part of the manuscript and should be discussed in more details.

---

## Round 0.2 · Minor Revisions

Dear Author,

Kindly fulfill reviewer's query and submit a revise version with a rebuttal.

Reviewer 1 ·

Basic reporting

Please see Additional Comments.

Experimental design

Please see Additional Comments.

Validity of the findings

Please see Additional Comments.

Additional comments

Thank the authors for responding to the comments. However, the authors did not seem to address one issue thoroughly:

As to the previous comment 2 ("To help readers to get a better understanding of the current review's novelty — how this review advances our previous knowledge about the CMTM family in tumors, it would be more informative to compare this manuscript with similar reviews (PMID: 32944388, PMID: 27356683, and PMID: 36792080) by summarizing the current review's strengths over others"), the authors answered in a general way but a more detailed response would be essential. To this end, please provide the comparison in a bullet point form. At the same time, please use several sentences to incorporate the comparison into the manuscript and list the sentences in the next response letter.

---

## Round 0.3 · accepted · Accept

It is a pleasure to accept your manuscript entitled " Research insights into the CMTM family: their roles in various tumors" in its current form for publication in PeerJ.